# Microstructural Design of Necklace-Like Fe_3_O_4_/Multiwall Carbon Nanotube (MWCNT) Composites with Enhanced Microwave Absorption Performance

**DOI:** 10.3390/ma14174783

**Published:** 2021-08-24

**Authors:** Mu Zhang, Sinan Song, Yamin Liu, Zaoxia Hou, Wenyi Tang, Shengnan Li

**Affiliations:** 1Key Laboratory for Anisotropy and Texture of Materials (Ministry of Education), School of Materials Science and Engineering, Northeastern University, Shenyang 110819, China; 1970391@stu.neu.edu.cn (S.S.); 2000500@stu.neu.edu.cn (Z.H.); 1770358@stu.neu.edu.cn (W.T.); 2Foshan Graduate School, Northeastern University, Foshan 528311, China; 3Comprehensive Technology Center of Foshan Customs, Foshan 528000, China; fsitc@163.com

**Keywords:** composite materials, carbon nanotube, microwave absorption, double loss mechanism, necklace-like structure, interfacial polarization

## Abstract

In order to manufacture microwave absorbers with strong attenuation abilities and that are light weight, in this paper, ferromagnetic carbon matrix composites were prepared by the composite of carbon nanotubes with adjustable dielectric constant and Fe_3_O_4_. Fe_3_O_4_/MWCNT composites with well-designed necklace-like structure and controllable size in the range of 100–400 nm have been successfully achieved by a simple solvent thermal method. A series of samples were prepared by changing experimental parameters. The microwave absorption characteristics of these samples were studied from the dielectric constant and magnetic permeability in two aspects. The electromagnetic absorption properties of the composites show obvious differences with different microsphere sizes, different microsphere density and different proportion of additives. When the solvothermal time is 15 h and the microsphere size is 400 nm, the reflection loss reaches −38 dB. The interfacial polarization caused by the unique structural design and good impedance matching produce composites that possess excellent electromagnetic loss ability.

## 1. Introduction

Carbon nanotubes-based composites materials have been widely accepted as microwave absorbers due to their inherent features such as low density and good conductivity. Therefore, they have received the extensive attention of researchers from various fields in recent years [1,2,3,4]. As is well known, carbon nanotubes are one of the important materials in the field of electromagnetic absorption [5,6,7,8]. It is well reported that the characteristics of MWCNTs as stealthy technology materials include a wide absorption spectrum and high absorption intensity, and they are light weight [9,10,11,12,13]. However, due to the intrinsic strong conductivity of carbon nanotubes [14,15,16,17], the real and imaginary parts of their complex permittivity are very large, but it is difficult to match with the free space, resulting in the electromagnetic wave not being able to enter the absorbent body but is instead reflected on its surface [18,19]. One method to overcome this problem is to combine magnetic materials with the composites. Thus, the real and imaginary parts of the complex dielectric constant of carbon nanotubes are reduced, and the matching between the carbon nanotubes and the air impedance is optimized. At the same time, adding magnetic material can also enhance the magnetic loss of the composite to the electromagnetic wave and improve the efficiency of microwave absorbing.

Among magnetic materials, ferrite has become a research hotspot because it has both dielectric loss and magnetic loss and low dielectric constant, which is conducive to impedance matching and electromagnetic wave entering the material. Fe_3_O_4_ is a kind of wave absorbing material that has been studied extensively at present. With the deepening of the research, Fe_3_O_4_ appears remarkable among absorbing materials because of its unique magnetic, strong spin polarization and semi-metal properties [20,21,22]. Fe_3_O_4_ and Carbon composites have attracted more and more attention. Deng et al. successfully synthesized HcFe_3_O_4_@C composites with the colloidal template method and pyrolysis method. The three-dimensional honeycomb structure effectively improves the interaction between the electromagnetic wave and the absorbing medium [23]. Huang et al. reported a strategy for preparing carbon microtubules (CMT)/Fe_3_O_4_ nanocomposites. The size of the prepared Fe_3_O_4_ nanoparticles is about 400 nm, the maximum reflection loss is about −40 dB, the effective absorption band width is about 4 GHz and the thickness is about 2.0 mm [24]. In summary, the composites of ferrite and carbon materials are amongst the most popular electromagnetic wave absorption materials at present, and the carbon-based iron oxide composite materials with special structure effectively promotes the improvement of the wave absorption performance. However, there are still some problems, which include the following: difficult preparation methods, the carbon material is completely coated and the conductive advantage is difficult to reflect. Therefore, we plan to design a unique necklace-like structure of Fe_3_O_4_ and carbon nanotubes composite absorbing material.

In this work, a Fe_3_O_4_/MWCNT composite absorbing materials with double dielectric loss and magnetic loss has been successfully prepared, and appropriate preparation methods have been explored in order to regulate its microstructure, improve the flexibility of its application, broaden its application range and further study the absorbing mechanism of composite nano-absorbing agent. The relationship between the microstructure and the absorbing properties of the composite nanometer absorbent was further analyzed.

## 2. Materials and Methods

### 2.1. Materials

Ferric chloride (FeCl_3_·6H_2_O, 99.0%) hexahydrate, sodium acetate (NaAC), ethylene glycol (EG), polyethylene glycol (PEG) and urea (CH_4_N_2_O) were obtained from Sinopharm Chemical Reagent Co. Ltd., Shanghai, China. The carbon nanotubes (≥95%) of Fe_3_O_4_/MWCNT used to prepare the necklace structure were purchased from Xianfeng Nanomaterial Technology Co. Ltd. (Nanjing, China). The above chemicals are of analytical grade and have not been further purified when used.

### 2.2. Functionalization of MWCNT with -COOH Groups

In a typical procedure, 200 mg MWCNT was added into a three-necked flask containing a mixture of concentrated nitric acid and concentrated sulfuric acid (volume ratio: 3:1). The mixed solution described above was dispersed by continuous strong ultrasound for 3 h. When the MWCNT is fully dispersed, the three-necked flask is placed in an oil bath, heated to 80 °C and kept at that constant temperature for 1 h. The sample was rinsed to neutral with a large amount of deionized water by pumping and filtration and dried at 60 °C for 8 h. The functionalized carbon nanotubes are accepted for subsequent use.

### 2.3. Synthesize Fe_3_O_4_/MWCNT Necklace-Like Structure

The amounts of 0.27 g (1 mmoL), 0.81 g (3 mmoL) and 1.35 g (5 mmoL) FeCl_3_·6H_2_O with different mass were dissolved in 40 mL EG solution, respectively. After the solution was fully dissolved, 200 mg functionalized MWCNT was weighed and added into the orange solution for continuous ultrasound for 3 h. The amount ofs 3.6 g NaAC and 1.0 g PEG were slowly added when MWCNT was fully dispersed and stirred at high speed for 30 min. The solution was transferred to a 100 mL reactor and stored at 200 °C for 10 h. It was then rinsed repeatedly with plenty of deionized water and ethanol. Vacuum drying was conducted at 60 °C for 8 h. The above steps were repeated without adding carbon nanotubes in order to make pure Fe_3_O_4_ clusters.

Next, in order to explore the influence of different particle size and density distribution on the wave absorption performance, we successfully prepared samples with increased distribution density and cluster average particle size of 100 nm, 200 nm, 300 nm and 400 nm by adjusting the experimental parameters.

It was difficult to adjust the size of the samples with special morphology in this paper by changing a single variable. In order to obtain samples with different particle sizes, the other three parameters were adjusted under the premise that the additional amount of MWCNT (200 mg) and PEG (1.0 g) and the heat treatment temperature (200 °C) remained unchanged. The adjustment variables of the four samples with different particle sizes are as follows: 100 nm of(FeCl_3_·6H_2_O (3 mmoL), NaAC (0.005 mmoL) and heat preservation time of 10 h); 200 nm (FeCl_3_·6H_2_O (1 mmoL), NaAC (0.04 mmoL) and heat preservation time of 5 h); 300 nm (FeCl_3_·6H_2_O (3 mmoL), NaAC (0.04 mmoL) and heat preservation time of 10 h); 400 nm (FeCl_3_·6H_2_O (5 mmoL), NaAC (0.04 mmoL) and heat preservation time of 15 h).

### 2.4. Characterization

The crystal phases of the Fe_3_O_4_/MWCNT nanomaterials were observed by using an X-ray diffractometer (XRD, Model Smartlab 9, Tokyo, Japan) with Cu Kα X-ray radiation. Scanning electron microscopy (SEM, JSM-7001F, Tokyo, Japan) and transmission electron microscopy (TEM, JEM-2100F, Tokyo, Japan) were performed to analyze the morphologies and structures of the Fe_3_O_4_/MWCNT nanomaterials. The elemental composition and chemical status of the samples were tested by X-ray Photoelectron Spectroscopy (XPS, Kratos, Manchester, UK).

### 2.5. Absorbing Performance Characterization

The sample powder was miscible with paraffin in a ratio of 3:7 at 90 °C. The above material is quickly poured into a hollow cylindrical die with an outer diameter of 7 mm and an inner diameter of 3 mm. It was tested by vector network analysis (VNA, Agilent N5234A, California, CA, USA) in the frequency range of 2 to 18 GHz.

## 3. Results and Discussion

We successfully synthesized the necklace-like Fe_3_O_4_/MWCNT heterostructure by a simple solvothermal method using functional carbon nanotubes as raw materials, and the synthesis strategy of Fe_3_O_4_/MWCNT composites is shown in Figure 1.

First, carbon nanotubes are fully dispersed in EG solution, and a large number of carboxyl groups on the side walls of MWCNT can adsorb Fe^3+^ ions in the FeCl_3_ solution through static electricity and bind firmly. In the solvothermal process, some Fe^3+^ ions were reduced to Fe^2+^ ions in situ and then coprecipitated into Fe_3_O_4_ crystals, thus realizing the position-selective modification of MWNTs with Fe_3_O_4_ nanocrystals. Due to the preferred orientation of the adjacent particles of the nanocrystals and the connection of these particles on the plane interface, the high surface energy decreases, which serves as the main driving force for realizing the agglomeration of the nanocrystals [25]. Dipole interactions among Fe_3_O_4_ nanocrystals also contribute to their directional aggregation. The synthesis process of Fe_3_O_4_ nanocrystals is divided into hydrolysis of metal ions and dehydration of metal hydrate. In this process, the glycol solvent and the metal precursor undergo a REDOX reaction. It is not only used as a polar solvent with high boiling points but also plays an important role in reducing Fe^3+^ from trivalent to bivalent. At the same time, compared with the aqueous solution, the non-aqueous solution of the system has fewer hydroxyl groups on the surface and higher viscosity, which can reduce the agglomeration rate of Fe_3_O_4_ nanocrystals and provide enough time for Fe_3_O_4_ nanocrystals to rotate to the low energy configuration interface. In addition, PEG can prevent the rapid growth of Fe_3_O_4_ crystals, which is conducive to orientation agglomeration. In this process, sodium acetate (NaAC) plays an important role in accelerating the formation of magnetite clusters. NaAC as a precipitator, through hydrolysis for forming alkaline conditions, promotes the hydrolysis of Fe^3+^ in the solution. Fe^3+^ is partially reduced to Fe^2+^ by ethylene glycol, which promotes the formation of Fe_3_O_4_ nanocrystals. Therefore, we completed the preparation of necklace-like composites.

The phase composition and structure of the prepared F_3_O_4_/MWCNT samples with different concentrations of Fe^3+^ and pure spherical Fe_3_O_4_ nanoclusters were studied by XRD, as shown in Figure 2. The diffraction peak at 2θ = 26° of the composite is different from that of the pure Fe_3_O_4_, which corresponds to the (002) crystal plane of the MWCNT (JCPDS No. 26-1077). It can be determined from the spectrogram that the prepared sample is a two-phase complex of Fe_3_O_4_ and carbon nanotubes. The diffraction peak of Fe_3_O_4_ (JCPDS No. 75-0033) is clear and strong without other impurity peaks, which indicates that the crystallinity and purity of Fe_3_O_4_ in the complex is high.

SEM and TEM were used to characterize the microscopic morphology of the samples, as shown in Figure 3 and Figure 4. The sample was composed of spherical nanoclusters and cylindrical carbon nanotubes, and the nanospheres wrapped the carbon nanotubes to form a necklace-like structure. SEM of Figure 3 clearly show that when other parameters remain unchanged, the number and distribution density of spherical Fe_3_O_4_ clusters obtained by adding different amounts of Fe^3+^ are significantly different. As can be observed from Figure 3a, when the added mass was 1 mmoL, the initial agglomeration had been formed. When the added amount of Fe^3+^ increased to 5 mmoL, the number of Fe_3_O_4_ microspheres increased, but the size change was minimal. It is not difficult to understand that the increase in Fe^3+^ concentration will produce more Fe_3_O_4_ nanocrystals and increase the Fe_3_O_4_ microsphere loading capacity on the surface of MWCNTs.

Based on the above successful preparation of Fe_3_O_4_/MWCNT composite materials, the experimental parameters were optimized to explore the experimental parameters that are beneficial to the wave absorption performance, and the SEM of four samples with different particle sizes were obtained, as shown in Figure 4.

In addition to different particle sizes, we can observe from Figure 4a–d that, as the average particle size increases from 100 nm to 400 nm, the loading capacity of the microspheres on the carbon nanotubes increases, the interface in the material increases and the necklace-like structure piles up with one another to form a network. It provides the conditions for improving the performance of wave absorption.

The Fe_3_O_4_/MWCNT composite structure was characterized by TEM, and the results were shown in Figure 5.

In the Figure 5a, the diameter of Fe_3_O_4_ microsphere is about 200–300 nm, and the diameter of MWCNT is about 30–40 nm. The necklace-like structure of the composite material is consistent with the SEM image. The HRTEM images of Figure 5b show crystal lattice spacing of 0.48 nm and 0.29 nm, which can be divided into (110) and (220) planes of Fe_3_O_4_, respectively. Furthermore, it is proved that the microspheres are composed of Fe_3_O_4_ nanocrystals. In order to obtain clearer and intuitive TEM characterization results, Fe_3_O_4_/MWCNT was embedded in the resin, the diamond knife of Leica ultra-thin slicer was used to cut the nanosheets with a thickness of 80 nm and the nanosheets were placed on a copper net for TEM characterization, as shown in Figure 5c. It is clearly shown that the sample consists of spherical nanoclusters formed by nanocrystalline agglomeration and cylindrical carbon nanotubes, and carbon nanotubes pass through the center or near the center of the cluster to form the structure we designed.

In order to further study the chemical composition and chemical bonds of the composite material, XPS characterization was carried out, as shown in Figure 6. All the binding energies were standardized by using C 1s at 284.8 eV as the reference. It can be clearly observed from the full spectrum in Figure 3a that there are three elements present, Fe, O and C, respectively. The two broad peaks located at 725.3 eV and 711.7 eV in the XPS map of Fe 2p in Figure 3b correspond to the Fe^3+^ characteristic peaks, and the binding energies centered at about 710.4 and 724.0 eV are characteristic peaks of Fe^2+^. The coexistence of the two valence states of Fe^3+^ and Fe^2+^ determines the composition of Fe_3_O_4_ [26]. In Figure 3c, the O 1s spectrum can be divided into three parts: Fe_3_O_4_, OH and H_2_O molecules at 529.3 eV, 531.4 eV and 532.8 eV, respectively [27]. The three peaks of C 1s spectrum at 284.8 eV, 286.3 eV and 288.6 eV correspond to C-C, C-O and C=O, respectively [28].

Through the above analysis, we believe that the necklace-like Fe_3_O_4_/MWCNT nanocomposites have been successfully prepared. Next, we compare and analyze the microwave absorbing performance of the composite materials prepared in this paper with pure Fe_3_O_4_. As we all know, the complex permittivity (ε = ε′ − jε″) and permeability (µ = µ′ − jµ″) of EM wave absorber play important roles in determining the reflection and transmission measurements [29]. Figure 7a shows the real and imaginary parts of the permittivity of Fe_3_O_4_/MWCNT nanocomposites in the range of 2–18 GHz. 

The ε′ value of Fe_3_O_4_/MWCNT composites with different Fe^3+^ additions fluctuates in the range of 10–55. The real and imaginary parts of the permittivity of the 3 mmoL and 5 mmoL additive amount samples did not fluctuate much with the increase in frequency, and a slight formant peak appeared near 17 GHz. The ε′ of 1 mmoL Fe^3+^ sample was higher and reached the maximum value at 17 GHz, which proved that the sample had better electromagnetic wave storage and polarization ability. The ε″ value is minimized at this frequency. This phenomenon shows that dielectric loss and electric dipole moment polarization are related. The μ′ value decreased slowly from 2 to 18 GHz, and the samples with different Fe^3+^ supplemental levels all reached the minimum value at 17 GHz. The minimum μ′ values of 1 mmoL Fe^3+^, 3 mmoL Fe^3+^ and 5 mmoL Fe^3+^ were 0.1, 0.6 and 1.0, respectively. With the increase in the sample concentration, the minimum μ′ values were significantly increased. When Fe^3+^ was added to 1 mmoL, µ″ reached the maximum value at 10 GHz and then decreased slowly. Similar to permittivity, µ″ increased first and then decreased (Figure 7c). All of these observations indicate that the Fe_3_O_4_/MWCNT necklace-like composites have dielectric loss and magnetic loss. In the range from 2 to 17 GHz, the value of tanδε varies near 0, while the value of tanδμ varies in the range of 0.2–1, indicating that the attenuation of electromagnetic wave is mainly due to the magnetic loss in this frequency range. In the range of 17–18 GHz, the tanδε is greater than the tanδμ, indicating that the dielectric loss is dominant in this band (Figure 7b,d).

In order to further explore the absorbing performance, we calculated the reflection loss (RL) of different thicknesses within the frequency range of 2~18 GHz according to Formula (1) and (2), where *Z_in_* is the normalized input impedance at the free space and material interface, *ε_r_* is the complex permittivity, *µ_r_* is the complex permeability of absorber, *f* is the frequency of EM wave in free space, d is the thickness of the absorber and c is the velocity of light in free space.
(1)$$Z_{in}=Z_{0}\sqrt{\mu_{r}/\varepsilon_{r}}\tan h\left[j\left(\frac{2\pi fd}{c}\right)\sqrt{\mu_{r}\varepsilon_{r}}\right]$$
(2)$$RL\left(dB\right)=20\log\left|\frac{Z_{in}-Z_{0}}{Z_{in}+Z_{0}}\right|$$

The data of the three samples were calculated, and 2.55 mm coating thickness was selected to carry out. The reflection loss (RL) change with frequency was drawn, as shown in Figure 8.

It is obvious that when the Fe^3+^ supplemental amount is 5 mmoL, the absorbing effect is better, and the peak value of reflection loss reaches −30 dB at near 9 GHz. With the increase in Fe^3+^ addition from 1 mmoL to 3 mmoL, the reflection loss increased from −13 dB to −30 dB; thus, the absorbing performance was significantly enhanced. The electromagnetic field produces space charge polarization and causes the dielectric loss to increase with the increase in Fe_3_O_4_ content. The real and imaginary parts of the dielectric constant both increase with the increase in Fe_3_O_4_ content, which is mainly attributed to the increase in the number of iron ions, the number of electric dipole and the contact area of the two phases in Fe_3_O_4_/MWCNT composites, as well as the enhancement of ion displacement, electric dipole orientation polarization and interface polarization.

The electromagnetic parameters of microsphere size from 100 nm to 400 nm were compared (Figure 9) in order to explore the influence of Fe_3_O_4_ microsphere size and distribution density on the wave-absorbing performance.

The ε′ of 100 nm samples fluctuated in the range of 30–50, which was significantly higher than that of other samples. As the proportion of Fe_3_O_4_ nanocrystals decreases, the electrical conductivity of carbon nanotubes is enhanced. The ε′ of 200 nm, 300 nm and 400 nm samples fluctuates in the range of 5–30, and the ε′ of 200 nm size samples at 8 GHz fluctuates to the minimum value of 5. The ε′ of the samples with sizes of 300 nm and 400 nm fluctuates little with frequency. With the increase in frequency, the ε“ value of the 100 nm sample decreases greatly from 2 GHz to 18 GHz and the ε″ value decreases from 32 to 6. In the range of 2–17 GHz, the ε″ of the samples with 200 nm increases first and then decreases, and it reaches the maximum value of 20 at 7 GHz. The ε″ of the 300 nm and 400 nm samples fluctuates little with frequency. Multiple resonance peaks appear in the imaginary part of the complex permeability of the sample, which is generated by the exchange resonance. The fluctuation range of μ″ and µ″ of the sample at 200 nm is relatively large, and the fluctuation range of µ″ with 100 nm, 300 nm and 400 nm is 0.8–1.2. The µ″ of 300 nm and 400 nm decreased from 0.1 to −0.2 with the increase in frequency. The fluctuation range of µ″ for the 100 nm sample is from 0.3 to 0.7 and increases slowly with the increase in frequency. However, the fluctuation range of the real part of the magnetic permeability is always around 1, which means that the sample has a small contribution to the magnetic loss. There are many formants in Figure 9c that belong to the natural formants of magnetic materials. However, the shift of the formant position is related to the composite of the material and the nanometer scale of Fe_3_O_4_. The dielectric loss tangent of the samples in this group is between 0 and 5, and the dielectric loss tangent of the samples with dense clusters of 300 nm and 400 nm is smaller than that of the other samples. This may be due to the selective modification of Fe_3_O_4_ nanocrystals on the surface of MWCNT, which reduces the dielectric polarization of the material. It would be beneficial to improve the impedance matching of the material, thus enhancing the microwave absorption performance of the material. For the magnetic loss tangent, the value of tanδμ decreases slowly with the increase in frequency in the samples with 300 nm and 400 nm.

The three-dimensional diagram and contour diagram of the relationship between the reflection loss, frequency and sample thickness are drawn, as shown in Figure 10.

With the increase in the size and distribution density of Fe_3_O_4_ microspheres, the absorbing effect is enhanced. When the size of the Fe_3_O_4_ microsphere is 100 nm, the peak value of reflection loss is about −5 dB, which cannot meet the requirements of absorbing performance. When the size of Fe_3_O_4_ microsphere is 200 nm, the reflection loss is −16 dB. At around 12 GHz, the peak reflection loss of the 300 nm sample is −25 dB, and the bandwidth greater than −10 dB is about 1.5 GHz. When the coating thickness is 1 mm near 9 GHz, the reflection loss peak of 400 nm sample reaches −38 dB, and there are two reflection loss peaks at 9.5 GHz and 11 GHz. The frequency range for losses greater than −10 dB is 2.5 GHz. With the increase in the size and number of Fe_3_O_4_ microspheres, effective absorption (RL < −10 dB) occurs at the frequency range of 4 GHz to 18 GHz and a thickness of 1.0 mm to 3.8 mm. The sample still shows excellent absorbing performance in the thin coating thickness.

Considering the excellent electrical conductivity of carbon nanotubes and that charge accumulation brings more spatial polarization, it is possible for charge transfer and electron polarization to occur at the two-phase interface, thus affecting the wave absorption performance to a certain extent. We calculated the local electric field distribution at the interface, as shown in Figure 11. The charges have accumulated at the interface under alternative electromagnetic wave due to the different work function of MWCNTs and Fe_3_O_4._ To make the Fermi level equal at the interface, the energy bands move and curve, resulting in the appearance of the charge barrier and trap at the interface. It cannot be ignored that collective movements of interface dipoles have also contributed to the microwave absorption. In addition to the polarization at the defect and the interface, the high impedance matching and the 3D mesh structure allow more electromagnetic waves enter the absorber (Figure 12), and all of them play an important role in improving the electromagnetic absorbing performance.

## 4. Conclusions

From the above observations, we successfully realized the simple hydrothermal method to prepare MWCNT/Fe_3_O_4_ composites with controllable morphology. The necklace-like structure composites in this paper effectively solved the problem of poor impedance matching between a single carbon material and a single ferrite material. In addition to the double loss mechanism, through the special structure design, the increase in the Fe_3_O_4_ microspheres load on the surface of carbon nanotubes brings multiple relaxation and good conduction, which also enhances the loss of electromagnetic waves to a certain extent. When the solvothermal time is 15 h and the average size of the microspheres is 400 nm, the optimal reflection loss can reach −38 dB. This provides reasonable ideas for the future research of absorbing materials.

## Figures and Tables

**Figure 1 materials-14-04783-f001:**
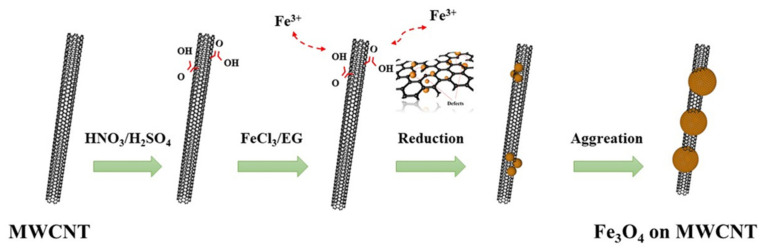
Synthesis strategy of Fe_3_O_4_/MWCNT heterostructure.

**Figure 2 materials-14-04783-f002:**
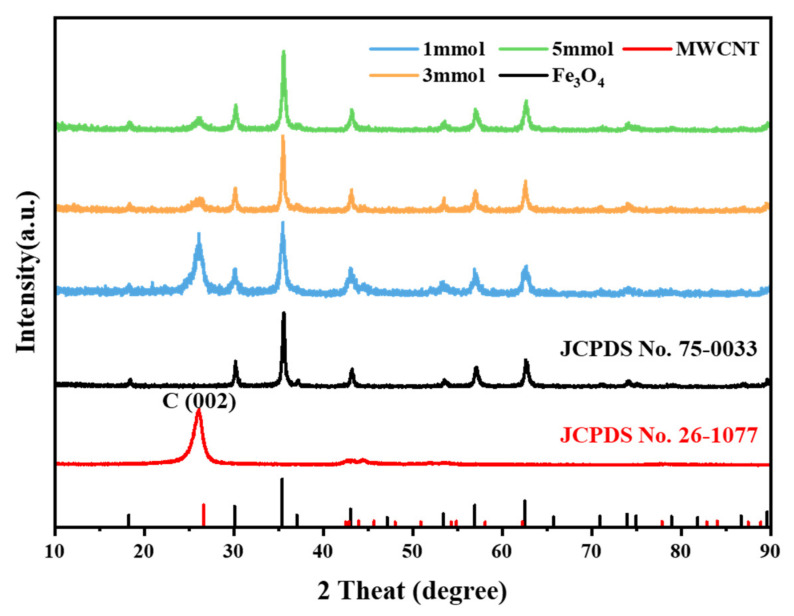
XRD patterns of Fe_3_O_4_, MWCNT and Fe_3_O_4/_MWCNT composite.

**Figure 3 materials-14-04783-f003:**

SEM images of Fe_3_O_4_/MWCNT composite with different Fe^3+^: (**a**) 1 mmoL, (**b**) 3 mmoL and (**c**) 5 mmoL.

**Figure 4 materials-14-04783-f004:**
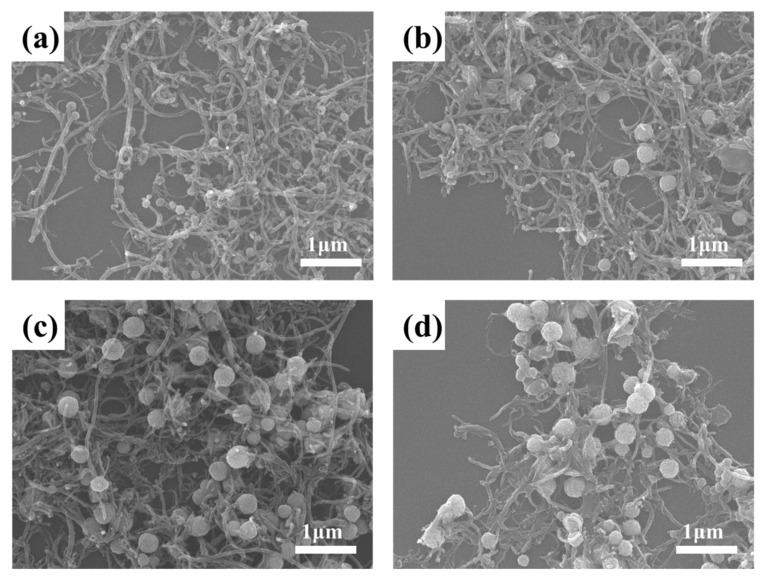
SEM images of Fe_3_O_4_/MWCNT composite with different size of clusters: (**a**) 100 nm, (**b**) 200 nm, (**c**) 300 nm and (**d**) 400 nm.

**Figure 5 materials-14-04783-f005:**
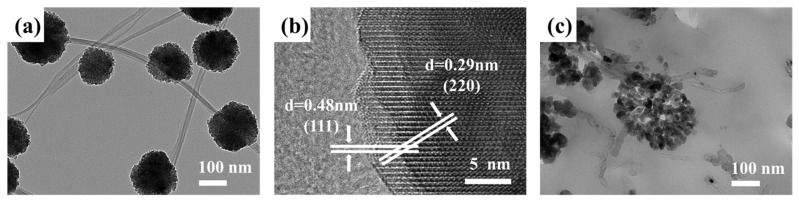
(**a**) TEM image of Fe_3_O_4_/MWCNT, (**b**) HRTEM image of Fe_3_O_4_/MWCNT and (**c**) TEM image of microtomed Fe_3_O_4_/MWCNT.

**Figure 6 materials-14-04783-f006:**
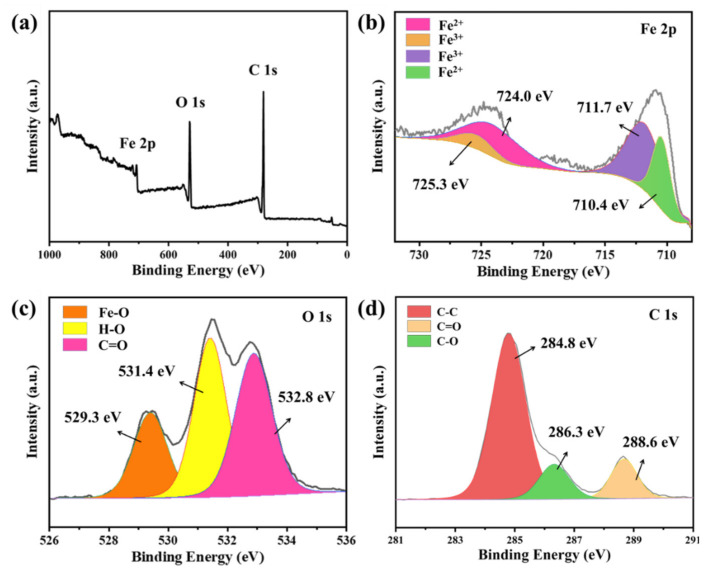
XPS spectrum of Fe_3_O_4_-MWCNT composites: (**a**) full spectrum, (**b**) Fe 2p, (**c**) O 1s and (**d**) C 1s.

**Figure 7 materials-14-04783-f007:**
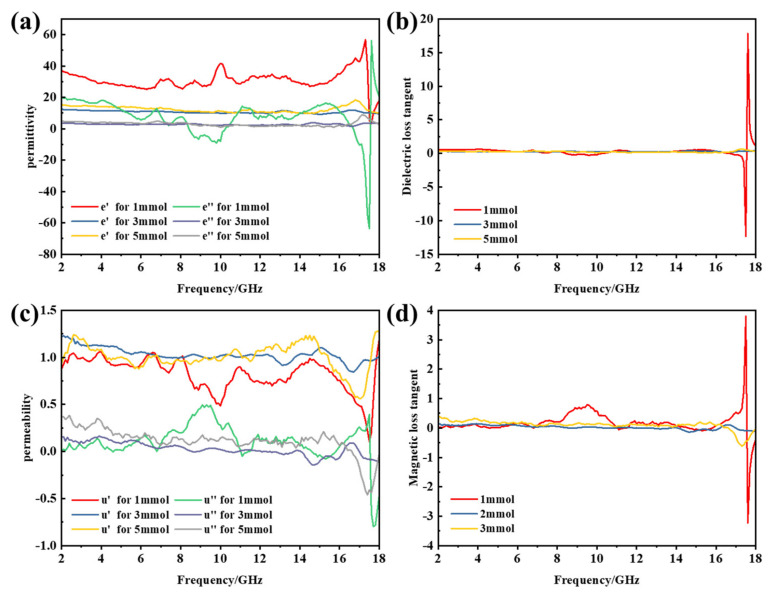
Permittivity and permeability spectra of Fe_3_O_4_/MWCNT composite materials with different Fe^3+^ concentrations: (**a**) real part and imaginary part of the permittivity, (**b**) dielectric loss factor, (**c**) real part and imaginary part of the permeability and (**d**) magnetic loss factor.

**Figure 8 materials-14-04783-f008:**
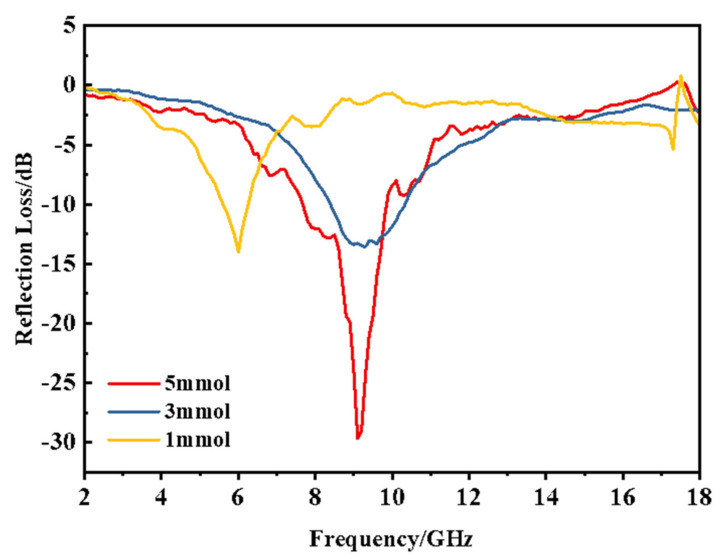
Reflection loss diagram of Fe_3_O_4_/MWCNT heterostructure with different Fe^3+^ concentrations.

**Figure 9 materials-14-04783-f009:**
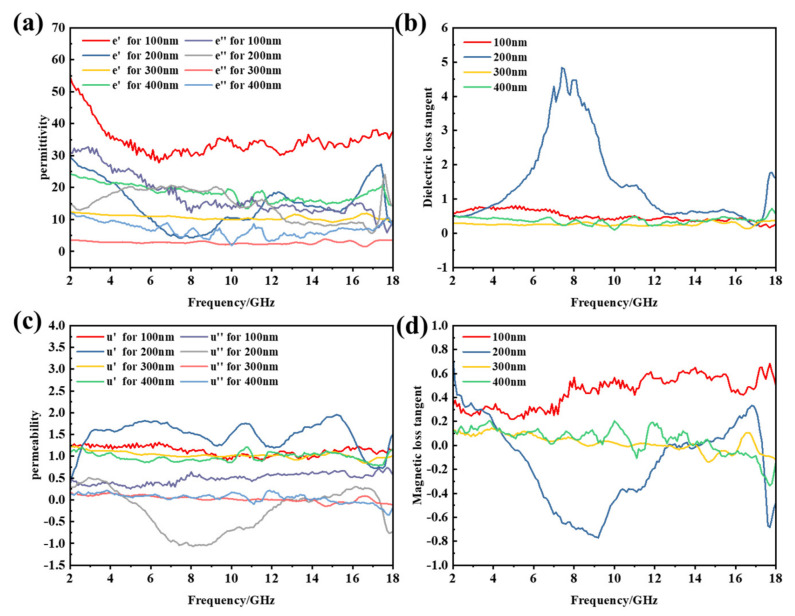
Permittivity and Permeability spectra of Fe_3_O_4_/MWCNT composite materials with different size of clusters: (**a**) real part and imaginary part of the permittivity, (**b**) dielectric loss factor, (**c**) real part and imaginary part of the permeability and (**d**) magnetic loss factor.

**Figure 10 materials-14-04783-f010:**
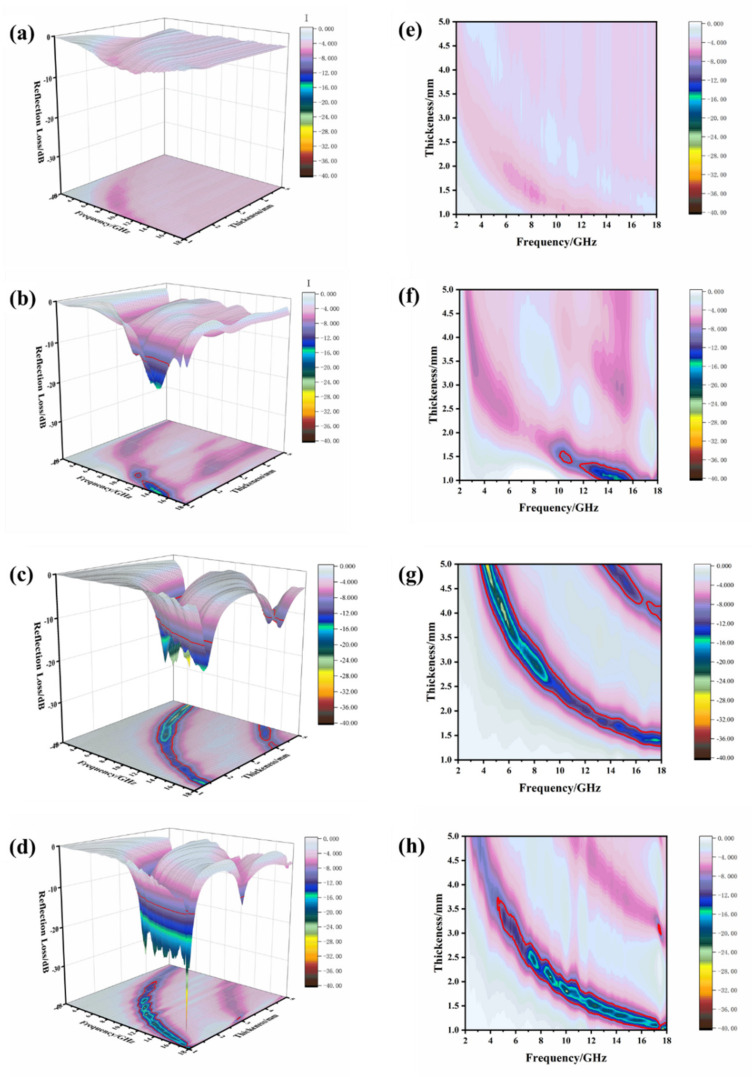
Three-dimensional diagrams and contour diagrams of the RL values of Fe_3_O_4_/MWCNT composite materials with different size of clusters: (**a**,**e**) 100 nm, (**b**,**f**) 200 nm, (**c**,**g**) 300 nm and (**d**,**h**) 400 nm.

**Figure 11 materials-14-04783-f011:**
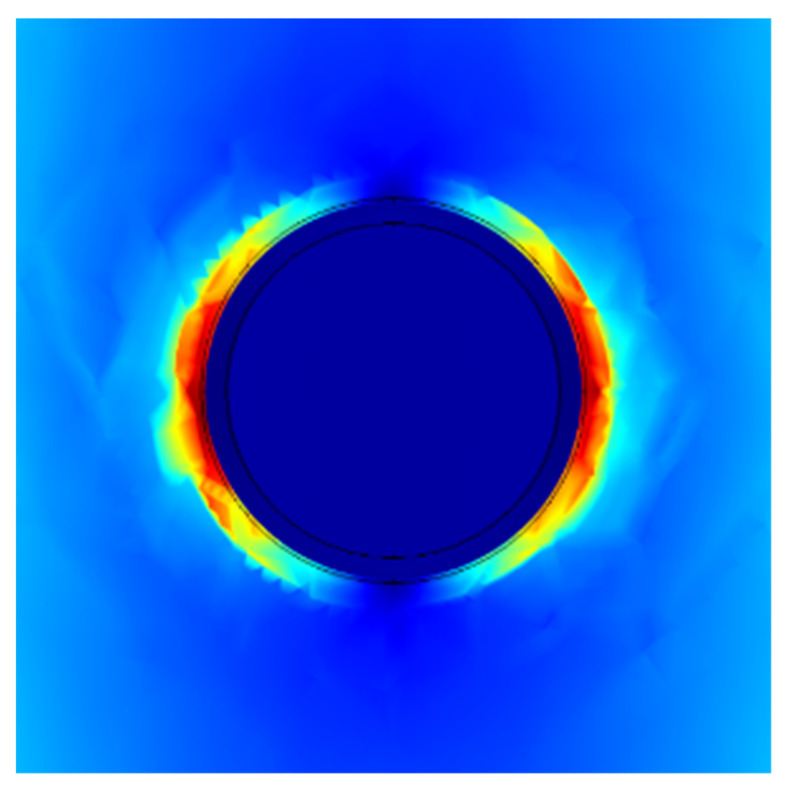
The distribution of electric field at the interface.

**Figure 12 materials-14-04783-f012:**
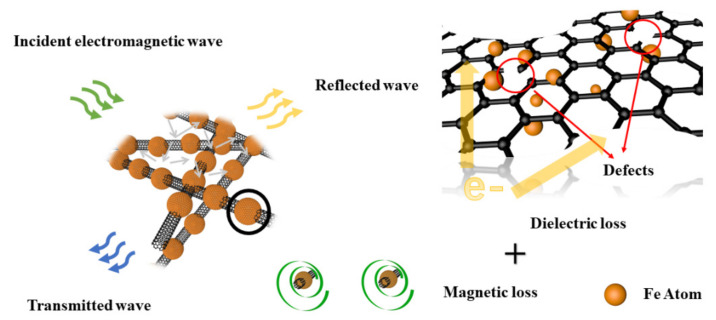
The absorbing mechanism of Fe_3_O_4_/MWCNT.

## Data Availability

The data presented in this study are available on request from the corresponding author. The data are not publicly available due to funder data retention policies.

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
