# Peer review of "Microstructural Design of Necklace-Like Fe3O4/Multiwall Carbon Nanotube (MWCNT) Composites with Enhanced Microwave Absorption Performance"

_materials, 2021, doi:10.3390/ma14174783_

Round 1

Reviewer 1 Report

In the manuscript submitted to “Materials”, Zhang et. al. reported necklace like Fe3O4/ MWCNT composites with controllable size leads to the enhanced microwave absorption performance. The reaction conditions to optimize the Fe3O3 nanocrystals were explored. The structural characteristics of the samples were further confirmed by XRD, SEM, XPS, and TEM analyses. Some more physical insights about the microstructure are beneficial and would enrich the work. This paper is an interesting work and well designed with good logic and structure arrangement. However, the manuscript has several fundamental problems, and it can be accepted after solving these basic problems.

1, In Figure 1, the authors should also provide the XRD pattern of MWCNT to better compare, and the references of both should be put in the figure or caption.

2, the authors claim “the experimental parameters were optimized to explore the experimental …”  to obtain the different sizes of Fe3O4 nanospheres, but the detailed strategy was not provided in the method or the corresponding discussion section.

3, For Figure 3, I suggest the authors should do Gaussian fitting statistics on the Fe3O4 particle size distribution.

4, Page 5, line 179, about the mechanism of the Synthesis strategy of Fe3O4/MWCNT heterostructure, I strongly suggest the authors should put this section at the beginning of the discussion because putting it here will make the reader's reading structure confusing.

5, another suggestion of this manuscript, the authors should enlarge the font of the vertical and horizontal coordinates of all the images, especially Figures 5, 7, and 8-10. The font is too small to read properly. There are some figures (e.g. Figures 5, 7, and 9) with four in one row, I suggest arranging them in two rows, please correct them accordingly.

Reviewer 2 Report

This manuscript reports on the development of necklace-like structures of Fe3O4 and multiwalled carbon nanotubes for use in microwave absorption. There have been numerous previous publications detailing the creation of microwave absorbers, but this report provides a unique view by the Fe3O4 growth on the CNT network. While this report demonstrates the creation of these, significant further editing and support is required before publication. Detailed comments and concerns about the manuscript are provided below.

Major notes:

  1. Please consider editing the manuscript for grammatical errors and syntax errors. Some portions were difficult to read.
  2. What is Figure 1 looking at? 1 mmol of Fe3+ in what? Is it CNTs? If so, what concentration? Why is the CNT peak strongest with 1 mmol of iron? How does this impact the final structure and absorption? All of this must be discussed before publication.
  3. How were the varying sizes of Fe3O4 synthesized? The article merely says “the experimental parameters were optimized “. What is the procedure? What Fe3+ concentration was used? Does this vary with different initial Fe3+ concentrations?
  4. Please quantify the concentrations of the CNTs and iron in all solutions. It may be possible that the change in wave absorption performance is at least partially due to concentration differences.
  5. Numerous images were unnecessary and confusing and should be removed – Figure4d, Figure 11 and 12.
  6. Please discuss each figure in order. The authors discussed figure 4d before figure 4c,
  7. The authors mention Figure 3 on line171 – should this be figure 5?
  8. Figures 7 and 9 are incredibly blurry. Please use a higher magnification image.
  9. Why are the permeability and permittivity trends so different in Figures 7 and 9? Shouldn’t at least one of the curves be identical? This indicates a significant amount of noise in the measurement. Please provide repeatability data for various devices as it appears that the conclusions of this report may be skewed by a singular result.
  10. Why is there a shift in reflectance loss between 1 and 3 mmol in Figure 8? Is this due to a change in thickness?
  11. It is difficult to assess the conclusions of this work as they appear unfinished. Given the strong dependence on size of the nanoparticle, the authors should perform a multitude of studies to optimize both size and concentration. It appears that the authors are modifying two factors simultaneously with the later portion of the study- size and density). Please perform a careful investigation to investigate each aspect individually to provide a thorough understanding of the dependance on the parameters.

Round 2

Reviewer 2 Report

The authors have adequately addressed all comments